# Biological Activity of Endophytic Fungi from the Roots of the Medicinal Plant *Vernonia anthelmintica*

**DOI:** 10.3390/microorganisms8040586

**Published:** 2020-04-17

**Authors:** Nigora Rustamova, Yanhua Gao, Yong Zhang, Abulimiti Yili

**Affiliations:** 1Key Laboratory of Plant Resources and Chemistry in Arid Regions, Xinjiang Technical Institute of Physics and Chemistry, Chinese Academy of Sciences, Urumqi 830011, China; n.rustamova@yahoo.com (N.R.); gaoyh@ms.xjb.ac.cn (Y.G.); 2University of Chinese Academy of Sciences, 19 A Yuquan Rd, Beijing 100049, China; 3State Key Laboratory of Microbial Metabolism, School of Life Sciences and Biotechnology, Shanghai Jiao Tong University, No. 800 Dongchuan Road, Minhang District, Shanghai 200240, China; yzhang2011@sjtu.edu.cn

**Keywords:** endophytic microorganisms, *Vernonia anthelmintica*, *Aspergillus* genus, GC/MS analysis, cytotoxic activity, antimicrobial activity

## Abstract

Endophytic fungi were first isolated from the fresh root of the Chinese medicinal plant *Vernonia anthelmintica* collected from the Hotan Prefecture within the Xinjiang Autonomous region of the People’s Republic of China. This plant has been used in Uyghur traditional medicine to treat vitiligo, a skin condition characterized by patches of the skin losing their pigment. In total, fifteen fungal strains were isolated. Among these, four endophytic fungi were identified by their DNA sequences and registered to GenBank with accession numbers. The isolates were identified as *Schizophyllum commune* XJA1, *Talaromyces* sp. XJA4, *Aspergillus* sp. XJA6, *Aspergillus terreus* XJA8. Ethyl acetate extracts of all fungal strains were used to quantify melanin content and to identify in vitro biological activity assays including antimicrobial, antioxidant, cytotoxic, antidiabetic and tyrosinase activity on B16 cells. Among the extracts of all four identified strains, the ethyl acetate extract of the *Aspergillus* sp. XJA6 was chosen for further characterization because it presented the highest biological activity against these tests. In addition, twenty four volatile compounds from the petroleum ether fraction were characterized by GC–MS.

## 1. Introduction

Plant endophytic microorganisms are symbiotic and live inside plants, and are further defined as bacteria, fungi, or viruses that spend all or part of their life cycle residing intercellularly/intracellularly within the host plant without causing overt negative effects [1,2]. Recent studies have isolated and identified endophytic microorganisms and the biologically active compounds they produce. Endophytic fungi, in particular, are a pivotal source of biologically active secondary metabolites [3]. Usually, secondary metabolites synthesized by endophytes are highly regulated and modulated at the genetic level; however, when the external environment is physiologically affected, these genes are further activated in response [4]. In addition, these compounds support the growth of host plants by increasing the resistance to external biotic and abiotic factors, and by enhancing resistance to insects and plant pathogens [5,6]. Many metabolites exhibit a range of biological activities and are important drivers of pharmaceutical innovation and discovery. Biologically active secondary metabolites include terpenoids [4,7], alkaloids [8], steroids [9], polyketides, quinones, lignans, and phenols [10]. These display various biological activities such as [9] antimalarial, anti-tubercular, antibacterial, antidiabetic [11], cytotoxicity [12], antioxidant [13], and acetylcholinesterase inhibition [14]. For example, secondary metabolites isolated from the fungus *Aspergillus niger* show moderate to strong antimicrobial activity against different test pathogens. [15]. Likewise, derived secondary metabolites from the fungus *Aspergillus flavipes* Y-62 displayed antimicrobial activity against the Gram-negative pathogens *Pseudomonas aeruginosa* and *Klebsiella pneumoniae* with equal minimum inhibitory concentrations (MICs) [2]. Xiao et al. [16] isolated compounds from the endophytic fungus *Aspergillus* sp. that are cytotoxic against the A549 cell line. The endophytic fungus *Aspergillus tamarii* from the roots of *Ficus carica* formed secondary metabolites with potential cytotoxic activity against human cancer cells such as MCF-7 and A549 [17]. Other natural products were isolated from a culture of the endophytic fungus *Diaporthe* sp. and displayed significant antioxidant activity [17]. Chemical constituents of the mangrove-derived fungus *Aspergillus* sp. were studied by Elissawy et al. [18] who identified the antiproliferative and very high cytotoxic activity of the produced natural compounds using Caco-2 cell lines. 

The medicinal plant *V. anthelmintica* belongs to the Asteraceae family. Recently, several chemical components were isolated from the seeds of *V. anthelmintica* and have demonstrated relevant biological activities, in particular with respect to the traditional Chinese medicinal use of the plant as a vermicide [19], as well as for treating skin diseases, including vitiligo, in traditional Uyghur medicine. These components define the main medicinal material in most preparations for treatment of vitiligo. Additional natural products isolated from seeds of *V. anthelmintica* include flavonoids [20,21], terpenoids, phenolic compounds and steroids [22]. However, constituents effective in treating vitiligo were not characterized. Therefore, we have isolated and systematically studied endophytic fungi from the root of the medicinal plant *V. anthelmintica* (Figure 1) growing in China. Fungal strains were cultured on a potato dextrose broth (PDB) medium. The culture was extracted with ethyl acetate to form crude extracts which were then tested for antimicrobial, cytotoxic, antioxidant, and antidiabetic activities, as well as screened for effects on melanin synthesis in murine B16 cells for vitiligo treatment.

## 2. Materials and Methods

### 2.1. Plant Collection

The Chinese traditional medicinal plant *V. anthelmintica* was collected from Hotan, within the Xinjiang Autonomous region, in October 2018. The fresh root of the plant was transported to the laboratory of Xinjiang Indigenous Medicinal Plants Resource Utilization, Xinjiang Technical Institute of Physics and Chemistry, Chinese Academy of Sciences. 

### 2.2. Isolation and Identification of Endophytic Fungi

The root systems were separated from the plant and the roots washed thoroughly with water [23]. The roots were then washed in sterile distilled water six times, soaked for 1–2 min in 70% ethanol followed by 5.25% sodium hypochlorite for 2–5 min, and again in 70% ethanol for 30–60 seconds before washing in distilled water. In order to obtain a root liquid, 2 g of root was crushed, 2–3 mL of sterile distilled water was added and the solution incubated for about 3 min. A total of 100 µL of root liquid was spread on sterile potato dextrose agar (PDA) medium containing chloramphenicol (100 μg/mL) as an antibacterial agent [19]. The PDA medium consisted of potato extract (4 g/L), dextrose (20 g/L), and agar (15 g/L) at pH 6.0. The medium was sterilized by autoclaving at 120 °C for 20 min. The fungal strains in the petri dishes were incubated at 28 °C for 7–10 days. After incubation, colonies with a different color, shape, and consistency were streaked on PDA plates and incubated for one week to ensure the purity of the isolates. Homogenous colonies of different colors, shapes, and sizes were used for DNA isolation [24]. Endophytic fungi were identified by their DNA gene sequences. 

### 2.3. Fermentation Medium

The endophytic fungi were cultivated in one liter of PDB liquid medium (dextrose 20 g/L, potato extract 4 g/L) and incubated at 28 °C on a shaker at 160 rpm for four weeks [24]. The culture was then filtered by vacuum filtration. Afterwards the culture filtrate was extracted with ethyl acetate (2 × 300 mL) in a separatory funnel (solvent–solvent extraction) [25]. The ethyl acetate extract was concentrated on a rotary evaporator. The crude extracts were dissolved in dimethyl sulfoxide (DMSO) and evaluated for their biological activities.

### 2.4. Antimicrobial Assay

The three pathogenic microorganisms used for the antimicrobial assays were the Gram-positive bacterium *Staphylococcus aureus* (ATCC6538), the Gram-negative bacterium *Escherichia coli* (ATCC11229) and the fungus *Candida albicans* (ATCC10231) [26]. The bacterial strains were plated on LB medium (4% glucose, 1% peptone) and the fungus in potato dextrose agar (PDA) (4 g potato extract, 20 g dextrose, agar 20 g, and 1 L water) by the well diffusion method, and incubated at 28 °C for 48 h in accordance with Santos et al. [24]. A total of 6 µL of cell-free suspension from endophytic fungi was added to these wells, and the plates were incubated at 37 °C for 24 h for the bacterial strains and 72 h for the fungus. Diameters of the microbial growth inhibition zones were measured to assess the antimicrobial activity of the samples. Ampicillin was obtained from Sigma Chemicals Co., and Amphotericin B. was purchased from AMRESCO LLC.

### 2.5. Melanin Content Assay

#### 2.5.1. Cell Culture

Murine B16 melanoma cell lines (B16F10) were obtained from CAS (Chinese Academy of Sciences, China). Cells were cultured in Dulbecco’s modified Eagle medium (DMEM, Gibco Life Technologies, Waltham, MA, USA) supplemented with 10% heat-inactivated fetal bovine serum (FBS), penicillin G (100 U/mL), and streptomycin (100 mg/mL) (Gibco-BRL, Grand Island, NY, USA) at 37 °C in a humidified atmosphere of 5% CO_2_.

#### 2.5.2. Melanin Measurement

The melanin content was determined in accordance with the procedure described previously [27,28,29,30]. B16 cells were seeded in a six-well plate at a concentration of 2 × 10^5^ cells per well and incubated for 24 h. The cultured B16 cells were further incubated in the absence or the presence of the sample for 48 h. Each well with cells was then washed twice with ice-cold PBS. Cells were then lysed as described previously [28]. A total of 150 µL aliquots of each lysate were put in a 96-well microplate and absorbance was read spectrophotometrically at 405 nm using a multi-plate reader. The protein content of each sample was determined by a BCA Protein Assay Kit (Biomed, Beijing, China). The melanin content was normalized to the cellular protein concentration. Melanin content was calculated according to Formula (1):(1)$$Melanin\;content\;\left(\%\right)=\frac{S}{B}\times 100$$
where *S* is the absorbance of the cells treated by the samples and *B* is the absorbance of the wells containing untreated cells. Each measurement was carried out in triplicate.

#### 2.5.3. Tyrosinase Activity Assay

Tyrosinase activity was performed using the method of Chao et al. [31]. First, the B16 cells were seeded in a six-well plate at a concentration of 2 × 10^5^ cells per well and incubated for 24 h. Test samples were then added to each well and incubated for 24 h. Cells were then washed twice with ice-cold PBS and lysed using 1% Triton X-100 solution containing 1% sodium deoxycholate for 30 min at −80 °C. The lysates were centrifuged at 12,000× *g* for 15 min. After quantification and adjustment of the protein concentration of the supernatants, 90 µL of supernatant was mixed and incubated with 10 µL of freshly prepared substrate solution (10 mM L-DOPA) in duplicate in a 96-well plate followed by incubation at 37 °C. Samples were then read at 490 nm. Results were calculated by the formula used for the calculation of melanin content.

### 2.6. Protein Tyrosine Phosphatase 1B (PTP1B) Inhibition Assay

The PTP1B Inhibitor was purchased from Merck specialties private limited. The PTP1B inhibitory activity of the cell-free supernatant of the endophytic fungal extract was carried out using *p*-nitrophenyl phosphate disodium salt (pNPP) as the substrate [32]. The crude extract of the fungal strains was dissolved in 50 mg/mL DMSO and pre-treated with the enzyme at room temperature for 5 min. One µL of a 0.115 mg/mL PTP1B protein solution was added to 178 µL of a solution of 20 mM HEPES buffer, 150 mM NaCl and 1 mM EDTA. The test and positive control samples were then added at a volume of 1 µL followed by addition of 20 µL of 35 mM substrate. The solution was mixed for 10 min. The plate was incubated in the dark for 30 min and the reaction terminated by addition of 10 µL of 3 M NaOH solution. The absorbance of the plate was read at 405 nm. The blank sample did not contain the enzyme. SpectraMax MD5 (USA Molecular Devices) was used to read the absorption of the plates. The inhibition rate (IR) of PTP1B by the samples was calculated using Formula (2):(2)$$\mathrm{IR}\;(\%)=\frac{PC-PS}{PC}\times 100$$
where *PC* is the absorbance of the control and *PS* is the absorbance of the sample.

### 2.7. Cytotoxic Activity (MTT Assay)

The [3-(4,5-dimethylthiazolyl)-diphenyl tetrazolium bromide] (MTT) assay was performed using the method of Bozorov et al. [33,34,35]. A stock solution of ethyl acetate extract of the fungus solution was prepared in 50 mg/mL DMSO. The anticancer assay was performed using MDA-MB-231 (breast cancer), Hela (cervical cancer), and HT-29 (colon cancer) cells. The positive control doxorubicin (DOX) was purchased from BBI Inc. (Shanghai, China), and the human cancer cell breast (MDA-MB-231), cervical (Hela), colon (HT-29) and human embryonic kidney (HEK-293) lines were obtained from Chinese Type Culture Collection, CAS (Shanghai, China). The cancer cell lines were separately seeded in 96-well plates at a density of 3–10^3^ cells per well followed by 24 h incubation at 37 °C, 95% humidity, and 5% CO_2_. After incubation, the cells were treated with 30 µM of test samples (crude extracts) and left for 48 h. Twenty µL of 5 mg/mL MTT was then added to each well and the plates were incubated at 37 °C for 4 h. Absorbance was read at 540 nm after removal of the supernatant and the addition of 200 µL DMSO to each well and the shaking of the multiwall plates for 10 min to completely dissolve the secondary metabolites. The inhibition rate was determined by Formula (3):(3)$$IR=\frac{\left(C-S\right)}{\left(C-B\right)}\times 100$$
where *C* is the OD of the control group, *S* is the OD of the experimental group, and *B* is the OD of the blank.

### 2.8. DPPH Radical Scavenging Activity Assay

The 2,2′-diphenyl-1-picrylhydrazyl (DPPH) test was carried out using the method of Rehebati et al. [36]. A 100-μL sample (concentration 50 mg/mL) was mixed with 100 μL of a freshly prepared 0.2 mM DPPH solution in each well of a 96-well microplate. The microplate was kept in the dark for 30 min at room temperature. The absorbance was recorded at 517 nm, and the inhibition rate was calculated. Vitamin C was used as a positive control. The inhibition rate (IR) of the positive control and samples was calculated using the following formula:(4)$$IR\left(\%\right)=1-\left(\frac{A_{0}-A_{1}}{A_{0}}\right)\times 100$$
where *A*_0_ is the absorbance of the blank and *A*_1_ is the absorbance of the test sample.

### 2.9. Gas Chromatography–Mass Spectrometry (GC–MS) Analysis

Gas Chromatography–Mass Spectrometry GC–MS analyses were conducted using the Agilent 6890 gas chromatograph-assisted Agilent 5973 mass detector (Agilent Technologies, USA). A capillary column (30 m × 0.25 mm internal diameter; CM Scientific, USA), coated with a 0.25-µM film of 5% (*v/v*) phenyl methyl siloxane, was used for separation. The oven temperature was set at 100 °C and held for 5 min, then increased to 145 °C at a rate of 5 °C/min, and held at this temperature for 25 min. The inlet temperature was 250 °C and the ionization source temperature was 280 °C. Data handling was performed using the Agilent ChemStation software (Agilent Technologies, USA) [37]. The compounds were identified on the basis of retention indices (RI) and by comparing the spectra with a stored MS library (W8N05ST and NIST08).

### 2.10. Statistical Analysis

Data were analyzed using GraphPad Prism using three replicate values in a side-by-side subcolumn analysis of variance (ANOVA), and Tukey’s test to determine statistical significance. A *p*-value of <0.05 was considered to be statistically significant. The correlation index was calculated using the Pearson coefficient (ρ).

## 3. Results and Discussion 

### 3.1. Isolation and Identification of Endophytic Fungi 

Fifteen fungal strains were isolated from the fresh roots of the medicinal plant *V. anthelmintica.* All isolated fungal strains were cultivated by PDB medium and extracted with ethyl acetate to produce crude extracts. Among these, only four crude extracts of the endophytic fungi (*S. commune* XJA1, *Talaromyces* sp. XJA4, *Aspergillus* sp. XJA6, and *A. terreus* XJA8) were selected for our study. The primary screening of these crude extracts showed satisfactory antimicrobial activity. All fungal strains were identified by DNA sequencing and registered in GenBank (accession numbers MK736352, MK736671, MK748459, and MK748458; www.ncbi.nlm.nih.gov). The respective isolates showed 98%–99.79% homology to *S. commune* XJA1, *Talaromyces* sp. XJA4, *Aspergillus* sp. XJA6, and *A. terreus* XJA8. Two isolates belong to the genus *Aspergillus*. 

### 3.2. Chemical Composition of the Endophyic Fungus Aspergillus *sp.* XJA6 Extract by GC–MS

The primary biological screening revealed that the endophytic strains *S. commune* XJA1, *Talaromyces* sp. XJA4, *Aspergillus* sp. XJA6, and *A. terreus* XJA8 displayed antimicrobial activity, with *Aspergillus* sp. XJA6 exhibiting the highest activity. In addition, the volatile chemical composition of the *Aspergillus* sp. XJA6 extract was studied by GC–MS analysis. The volatile compounds of petroleum ether (1.D) and petroleum ether–ethyl acetate fractions at a ratio 80:1 (2.D) from *Aspergillus* sp. XJA6 crude extract were separated using column chromatography. This analysis revealed thirteen components in the **1.D** fraction (Table 1), the major components being dibutyl phthalate (46.57%), bis (2-ethylhexyl) phthalate (14.85%), diisobutyl phthalate (5.98%), and bis (2-ethylhexyl) adipate (4.21%). In addition, eleven components were found in the **2.D** fraction (Table 2) with the principal components being undecanoic acid (27.33%), palmitic acid (11.63%), stearic acid (5.31%) and di-*sec*-butyl phthalate (3.55%). It is notable that some volatile components were found in both fractions. Although compounds such as di-*sec*-butyl phthalate, methyl 14-methylhexadecanoate, bis (2-ethylhexyl) phthalate, and methyl 12-methyltetradecanoate were found in the both fractions, the amounts of these components differed between fractions **1.D** and **2.D**. 

### 3.3. Antimicrobial Activity of Endophytic Fungi 

A part of the present study focused on the systematic screening of crude extracts from endophytic fungi. The aerial parts of the *V. anthelmintica* are understood to have potential as adjuncts to the antimicrobial arsenal in herbal medicine. Isolated secondary metabolites from this medicinal plant showed promising antibacterial activities against several pathogenic microorganisms [38,39]. Our work is the first study of antimicrobial inhibition of extracts from endophytic fungi of the *V. anthelmintica* plant, and we have also studied extracts of the poorly characterized *V. anthelmintica* root. In this regard, we provide here the antimicrobial properties of the selected endophytic strains of *S. commune* XJA1, *Talaromyces* sp. XJA4, *Aspergillus* sp. XJA6, and *A. terreus* XJA8. The antimicrobial properties of the fungal crude extracts were estimated using three pathogenic microbes, *S. aureus* (ATCC6538) (Gram-positive bacteria), *E. coli* (ATCC11229) (Gram-negative bacteria) and *C. albicans* (ATCC10231) (fungi) (Table 3). The crude extract of *Aspergillus* sp. XJA6 (ethyl acetate extract derived from the filtrate of PDB culture) exhibited strong (23 mm zone of inhibition (ZOI)) antifungal activity against the *C. albicans* strain, as well as moderateantibacterial inhibition (11 and 8 mm ZOI) of *S. aureus* and *E. coli* strains. Extracts of *A. terreus* XJA8 showed greater antifungal activity to *C. albicans* (20 mm ZOI) and antibacterial activity against *S. aureus* and *E. coli,* with inhibition zones of 16 and 12 mm, respectively. The crude extracts of *S. commune* and *Talaromyces* sp. XJA4 showed moderate growth inhibition against *S. aureus, E. coli,* and *C. albicans* (12, 9.5, 11 mm, and 11, 9, 9 mm ZOI, respectively).

Among the prepared crude extracts of the endophytic strains, *A. terreus* XJA8 exhibited significant inhibition zones for all pathogenic microorganisms, while *Aspergillus* sp. XJA6 extract displayed significant antifungal activity against *C. albicans* (Figure 2). These observations confirm the antimicrobial behavior of the *Aspergillus* species. Previously, the antimicrobial activity of the endophytic fungal strain *Aspergillus* sp. ASCLA from leaf tissues of the medicinal plant *Callistemon subulatus* was investigated against *S. aureus*, *Pseudomonas aeruginosa*, *C. albicans*, and *Saccharomyces cerevisiae* and exhibited moderate to high activity [40]. In our case the *Aspergillus* strain was isolated from the root of this herbal plant.

As noted above, natural compounds isolated from *V. anthelmintica* exerted antibacterial properties against various pathogenic microbial strains. Steroid derivatives obtained from *V. anthelmintica*, for example, exhibited antibacterial activities against *B. cereus*, *S. aureus*, *B. subtilis* and *E. coli*, with MICs ranging from 3.15 to 15.5 μg/mL [38]. Ethyl acetate extracts studied in the present work possibly contain secondary metabolites such as steroids, terpenes or terpenoids, which may be responsible for the antimicrobial activities.

### 3.4. Effect of Crude Extracts of Endophytic Fungi Derived from V. anthelmintica Root on Melanin Content Assay and Tyrosinase Activity in B16 cells

It was previously reported that secondary metabolites from the seeds of *V. anthelmintica* demonstrated effects on melanin synthesis in B16 melanoma cells [20]. Recently we isolated the entophytic bacterium *Pantoea ananatis* obtained from *V. anthelmintica* root. The crude extracts of the ethyl acetate fraction formed by *P. ananatis* influenced melanin synthesis in murine B16 cells [27]. Moreover, seven alkaloids such as 1*H*-indol-7-ol, tryptophol, 3-indolepropionic acid, tryptophan, 3,3-di(1*H*-indol-3-yl)propane-1,2-diol, cyclo(L-Pro-L-Tyr) and cyclo[L-(4-hydroxyprolinyl)-L-leucine, along with one dihydrocinnamic acid were also derived from the entophytic bacterium *P. ananatis* crude extract. Note also that the secondary metabolite cyclo(L-Pro-L-Tyr) showed the strongest effect on melanin synthesis in murine B16 cells [27]. Continuing our previous research on endophytes of *V. anthelmintica* root, we have now investigated the effect of the selected crude extracts (*S. commune* XJA1, *Talaromyces* sp. XJA4, *Aspergillus* sp. XJA6, and *A. terreus* XJA8) of endophytic fungi produced from *V. anthelmintica* root on melanin synthesis in murine B16 cells. The crude extracts of endophytic fungi were evaluated for their effects on the synthesis of melanin pigments and intracellular tyrosinase activity in B16 cells. This method was carried out as described by Niu et al. [31] and Rustamova et al. [27]. After 48 h of treatment with crude extracts at 50 µg/mL, melanin content in B16 cells was determined (Figure 3).

The crude extract of the ethyl acetate fraction from endophytic fungi increased melanin synthesis in murine B16 cells in a dose-dependent manner, and this activity was comparable to that of the reference drug 8-methoxypsoralen (8-MOP) [30]. The crude extracts of fungal strains *A. terreus* XJA8 and *Aspergillus* sp. XJA6 strongly increased melanin synthesis by 177.92% and 182.45% at 50 µg/mL, while 8-MOP increased synthesis by only 133.34%. The crude extracts of *Talaromyces* sp. XJA4 (50 µg/mL, 118.16%) and *S. commune* XJA1 (50 µg/mL, 119.02%) demonstrated a similar influence on melanin synthesis in B16 cells. The crude extracts of *A. terreus* XJA8 and *Aspergillus* sp. XJA6 were selected for further investigation due to their strong melanin synthesis and tyrosinase activity in murine B16 cells (Figure 3). Strains *A. terreus* XJA8 (1 µg/mL, 95.58%; 10 µg/mL, 102.07% and 50 µg/mL, 165.46%), and *Aspergillus* sp. XJA6 (1 µg/mL, 92.86%; 10 µg/mL, 131.64% and 50 µg/mL, 183.32%; 8-MOP 50 µg/mL, 142.86%) also increased melanin synthesis in a dose-dependent manner. The tyrosinase activities of these strains in B16 cells were also determined, and treatment of ethyl acetate crude extracts exhibited a similarly increased response (Figure 4). Treatment with the selected crude extracts of endophytic fungi at concentrations from 1 to 50 µg/mL resulted in intensified intracellular tyrosinase activity (*Aspergillus* sp. XJA6, 1 µg/mL, 131.49%; 10 µg/mL, 125.22%; 50 µg/mL, 159.94%; 8-MOP 50 µM, 133.25%). The crude extracts were tested on melanin synthesis in murine B16 cells, and synergistic effects (effects of non-isolated secondary metabolites in the crude extract) showed that secondary metabolites influenced the activity on melanin synthesis compared to that of 8-MOP. Further separation and purification of the compounds is needed to determine which secondary metabolites (in crude extracts) influence activity, and our laboratory is actively pursuing these. Terpenes isolated from the seeds of *V. anthelmintica* showed melanin production in B16 cells [41].

### 3.5. Antidiabetic Activity (PTP1B Assay)

PTP1B plays an important role in the negative regulation of insulin signaling. Crude extracts of the endophytes and plants or their derived secondary metabolites inhibited PTP1B activity [42,43,44]. Crude extracts of the endophytic strains have not been sufficiently explored with regard to PTP1B inhibition activity, active secondary metabolites produced by endophytic fungi, or identification of bacterial strains exhibiting PTP1B activity. For example, novel guignardins isolated from cultures of the endophytic fungus *Guignardia* sp. KcF8, itself from the mangrove *Kandelia candel*, have been presented a new class of PTP1B inhibitors [45]. In 2019, Gao et al. reported PTP1B inhibitory activity from a new polyketide compound isolated from cultures of the endophytic fungus *Pestalotiopsis neglecta* [46]. In our study, we also investigated PTP1B inhibitory activity of crude extracts from *S. commune* XJA1, *Talaromyces* sp. XJA4, *Aspergillus* sp. XJA6, and *A. terreus* XJA8. The extracts were screened in vitro for antidiabetic activity by PTP1B enzyme inhibition assay. The crude extracts of *Aspergillus* sp. XJA6 and *Talaromyces* sp. XJA4 exhibited significant inhibition of PTP1B with IC_50_ values of 5.662 ± 1.099 μg/mL and 4.789 ± 1.222 μg/mL. In addition, crude extracts of the endophytic fungi *A. terreus* XJA8 and *S. commune* XJA1 moderately inhibited PTP1B (IC_50_ 23.439 ± 0.734 μg/mL and 11.964±0.484 μg/mL, Table 4). Note, however, that the crude extract of *A. terreus* XJA8 had only a modest influence on PTP1B enzyme inhibition, although this crude extract was more effective in the bioassays described above (antimicrobial and melanin synthesis in murine B16 cells). These observations and synergistic aspects of the crude extract components suggest that isolation of the appropriate secondary metabolites obviates the need to test PTP1B enzyme inhibition activity. 

### 3.6. Cytotoxic Activity of Ethyl Acetate Extracts Obtained from Fungal Endophytes

Literature reviews suggest that endophytic fungi and their derived secondary metabolites exhibit potential activity against specific cancer cell lines [47]. Anticancer properties are generally associated with the cytotoxicity of the secondary metabolites present in the endophytic fungi [48]. Many endophytic strains synthesizing anticancer agents were classified taxonomically and species of the genus *Aspergillus* were found to produce anticancer compounds [48,49]. In our case, among the four selected crude extracts of the endophytic fungi (*S. commune* XJA1, *Talaromyces* sp. XJA4, *Aspergillus* sp. XJA6, and *A. terreus* XJA8), two species are members of the *Aspergillus* genus. The MTT assay was used to assess the cytotoxic activity of these samples. Crude extract of fungi were measured against the following human cancer cell lines: HT-29 (colon cancer), MDA-MB-231 (breast cancer), and Hela (cervical cancer) (Table 5). The reference drug DOX was used as positive control. When tested on Hela and HT-29 lines, the ethyl acetate crude extracts of endophytic fungi *Aspergillus* sp. XJA6 and *A. terreus* XJA8 showed strong activity with IC_50_ values of 9.99 ± 0.8 μg/mL and 5.73 ± 0.6 μg/mL. IC_50_ values of extracts of *Aspergillus* sp. XJA6 and *Talaromyces* sp. XJA4 against the HT-29 cell line were 19.31 ± 0.8 μg/mL and 90.43 ± 0.01 μg/mL. Only the crude extract of *S. commune* XJA1 displayed no activity. When screened on breast cancer cells, extracts of *Aspergillus* sp. XJA6 and *A. terreus* XJA8 strain showed moderate activity (IC_50_ 33.55 ± 0.1 μg/mL and 56.3 ± 0.6 μg/mL). Crude extracts of *A. terreus* XJA8, *Talaromyces* sp. XJA4 and *S. commune* XJA1 IC_50_ activities on Hela cells were 24.69 ± 0.2.85 μg/mL, 46 ± 0.3 μg/mL and 89.8 ± 0.4 μg/mL, respectively. 

Crude extracts of *Aspergillus* sp. XJA6 and *A. terreus* XJA8, which exhibited lower IC_50_ values on human Hela and HT-29 cell lines, were selected for screening against normal human embryonic kidney (HEK-293) cells. The selectivity index (SI) was calculated using Formula (5):(5)$$\mathrm{SI}=\frac{\;{\mathrm{IC}}_{50}\;\mathrm{of}\;\mathrm{selected}\;\mathrm{samples}\;\mathrm{against}\;\mathrm{HEK}-293\;\mathrm{cells}}{{\mathrm{IC}}_{50}\;\mathrm{of}\;\mathrm{selected}\;\mathrm{samples}\;\mathrm{against}\;\mathrm{cancer}\;\mathrm{cells}}$$

Our results show only a modest SI value for *Aspergillus* sp. XJA6 (SI = 9.3, between Hela and HEK-293 cells) compared with known anticancer drugs (Table 6). However *A. terreus* XJA8 displayed an SI value of 13.8 between HT-29 and HEK-293 cells, warranting further screening.

Some secondary metabolites from *V. anthelmintica*, in particular the known sesquiterpene lactones, exhibited significant antiproliferative activities against the cancer cell lines A549, Hela and MDA-MB-231, displaying IC_50_ values up to 1.00 μM [50]. Further studies are needed to identify active natural compounds potentially useful in the discovery of novel cytotoxic drugs. 

### 3.7. DPPH Radical Scavenging Assay

Antioxidant properties of endophytes have been assayed using the DPPH radical scavenging assay. Liu et al. [51] evaluated antioxidant activities and chemical composition of endophytic *Xylaria* sp. from *Ginkgo biloba*. Cui et al. [52] investigated the antioxidant activity of secondary metabolites from *Aspergillus ochraceus*, an endophytic fungus produced by the marine brown alga *Sargassum kjellmanianum*. These investigations have provided important information in our studies using the DPPH radical scavenging assay. In the current study, the antioxidant capacities of the selected crude extracts of endophytic fungi tested were in 0.005 μg/mL, and concentrations were evaluated by the most commonly used antioxidant assay, DPPH (Table 7). The *S. commune* XJA1 extract showed moderate antioxidant activity with an IC_50_ value of 55.21 ± 0.3 μg/mL. *Talaromyces* sp. XJA4 and *A. terreus* XJA8 extracts showed low inhibitions of 171.78 ± 8.06 and 215.838 ± 7.25 μg/mL, respectively. Extracts of *Aspergillus* sp. XJA6 did not display any activity. Crude extracts of *Aspergillus* also demonstrated no activity. 

## 4. Conclusions 

In the present study, the biological activities of endophytic fungi isolated from *V. anthelmintica*, grown in Hotan, People’s Republic of China, are demonstrated. Four fungal isolates were studied: two *Aspergillus* sp., one from *Talaromyces* sp. and the fourth from *Schizophyllum* sp. Ethyl acetate extracts of endophytic fungi exhibited several biological activities, including antimicrobial, cytotoxic, antioxidant, antidiabetic and melanin content, and tyrosinase activity on murine B16 cells. The *A.*
*terreus* XJA8 and *Aspergillus* sp. XJA6 showed strong activity in melanin synthesis. Moreover, these strains, such as *Aspergillus* sp. XJA6 and *A. terreus* XJA8 strain, showed activity against the following human cancer cell lines: Hela (cervical cancer), HT-29 (colon cancer) with IC_50_ of 9.99 ± 0.8 μg/mL and 5.73 ± 0.6 μg/mL, respectively. *Aspergillus* sp. XJA6 showed higher biological activity than the other strains. Twenty-four volatile compounds from the petroleum ether fraction were characterized by GC–MS in our study, but others, particularly those from the ethyl acetate fractions, require further investigation to determine the components with potential in future drug development. Our laboratory is continuing to pursue this research. 

## Figures and Tables

**Figure 1 microorganisms-08-00586-f001:**
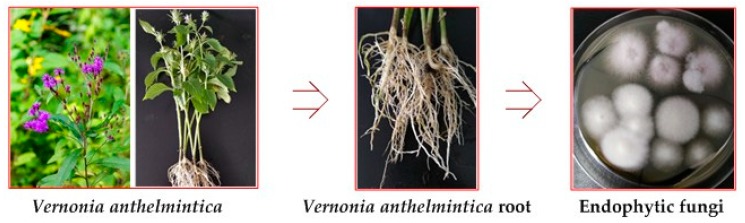
Main approach used in the present study.

**Figure 2 microorganisms-08-00586-f002:**
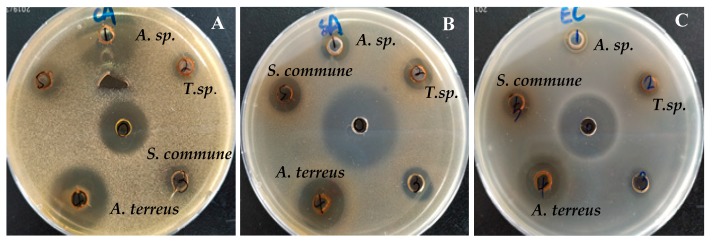
Antimicrobial activity of fungal endophytes from *V. anthelmintica* against some pathogenic fungus *C. albicans* (**A**) and bacteria *S. aureus* (**B**) and *E. coli* (**C**).

**Figure 3 microorganisms-08-00586-f003:**
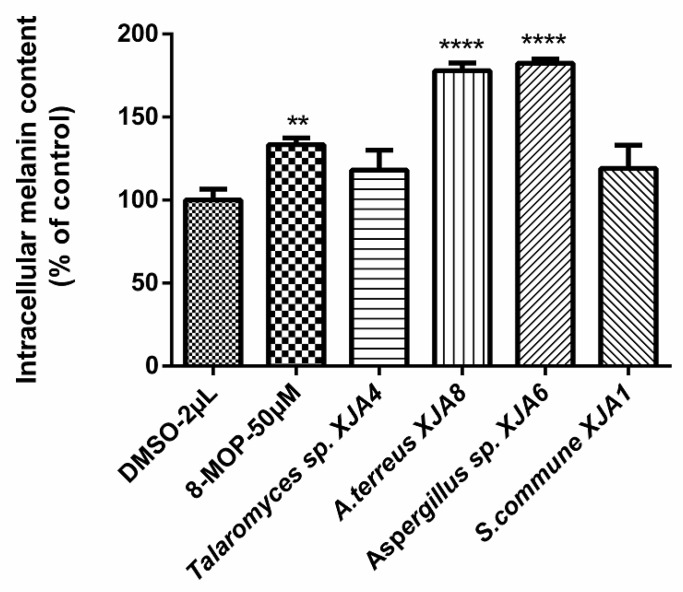
Effect of secondary metabolites of endophytic fungi on melanin production in B16 cells, (** *p* value ˂ 0.01, while **** *p* values ˂ 0.0001, compared with the dimethyl sulfoxide (DMSO)-treated control).

**Figure 4 microorganisms-08-00586-f004:**
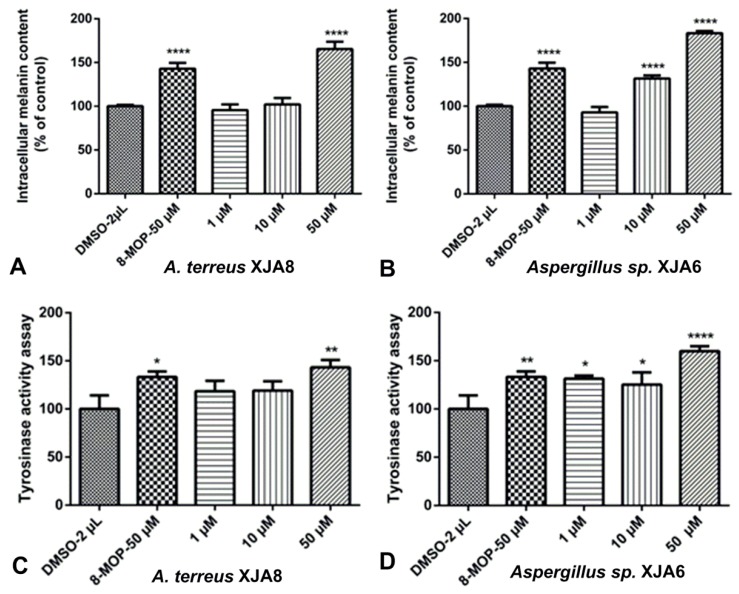
Stimulation of melanin content assay (**A**,**B**) and tyrosinase activity of B16 cells (**C**,**D**) by fungal strain crude extracts at different concentrations of *Aspergillus* sp. XJA6 and *A. terreus* XJA8. (All **** *p* values for melanin content and tyrosinase activity are ≤0.0001, compared with the DMSO-treated control. All * *p*-values for tyrosinase activity are ˂0.05, while ** *p*-values for tyrosinase activity are ˂0.01, compared with the DMSO-treated control).

**Table 1 microorganisms-08-00586-t001:** The chemical composition of petroleum ether (1.D).

No.	Composition	RT/min	Quantity(%)
1	2-(1-Methylcyclopropyl) aniline	17.435	0.76
2	Methyl 12-methyltetradecanoate	18.795	0.84
3	Diisobutyl phthalate	19.857	5.98
4	Methyl 14-methylpentadecanoate	20.443	1.00
5	Di-sec-butyl phthalate	20.850	0.25
6	Dibutyl phthalate	21.139	46.57
7	Methyl 14-methylhexadecanoate	21.360	0.75
8	(9Z,15Z)-Methyl 9,15-linoleate	22.473	0.67
9	Methyl oleate	22.533	0.71
10	Bis(2-ethylhexyl) adipate	25.752	4.21
11	Bis(2-ethylhexyl) phthalate	27.307	14.85
12	1,4-Benzenedicarboxylic acid	29.048	1.75
13	Bis(2-ethylhexyl) sebacate	29.550	0.82

**Table 2 microorganisms-08-00586-t002:** The chemical composition of petroleum ether–ethyl acetate (2.D).

No.	Composition	RT/min	Quantity(%)
1	Methyl 13-methyltetradecanoate	18.650	0.70
2	Methyl 12-methyltetradecanoate	18.761	1.86
3	Methyl 14-methylhexadecanoate	21.360	1.11
4	Undecanoic acid	19.13/21.73 ^a^	27.33
5	Oleic Acid	20.604	0.85
6	Palmitic acid	20.859	11.63
7	Di-*sec*-butyl phthalate	20.927	3.55
8	Methyl 14-methylhexadecanoate	21.284	1.65
9	Methyl (13Z)-octadecenoate	22.499	1.08
10	Stearic acid	23.170	5.31
11	Bis(2-ethylhexyl) phthalate	27.197	2.94

^a^ RT at two values.

**Table 3 microorganisms-08-00586-t003:** Antimicrobial activity of fungal endophytes from *V. anthelmintica* against some pathogenic bacteria and fungi.

Samples	Sample Concentration (mg/mL)	Sample Amount (µL)	*C. albicans*(ZOI)	*S. aureus*(ZOI)	*E.coli* (ZOI)
Ampicillin sodium salt	10	5			20
Ampicillin sodium salt	1	5		27	
Amphotericin B	5	20	17		
*Aspergillus* sp. XJA6	50	20	23	11	8
*Talaromyces* sp. XJA4	50	20	9	11	9
*A. terreus* XJA8	50	20	20	16	12
*S. commune* XJA1		20	11	12	9.5

**Table 4 microorganisms-08-00586-t004:** Antidiabetic activities of ethyl acetate crude extracts of selected endophytic fungi.

Sample	IC_50_ (μg/mL)
*Aspergillus* sp. XJA6	5.662 ± 1.099
*Talaromyces* sp. XJA4	4.789 ± 1.222
*A. terreus* XJA8	23.439 ± 0.734
*S.commune* XJA1	11.964 ± 0.484
PTP1B	1.46 ± 0.40

**Table 5 microorganisms-08-00586-t005:** Cytotoxic activity of ethyl acetate extracts obtained from fungal endophytes on human cancer cell lines HT-29, MDA-MB-231 and Hela.

Samples	Cell Lines		
	IC_50_ (μg/mL)		
	HT-29	MDA-MB-231	Hela
*Aspergillus* sp. XJA6	19.31 ± 0.8	33.55 ± 0.1	9.99 ± 0.8
*A. terreus* XJA8	5.73 ± 0.6	56.3 ± 0.6	24.69 ± 0.2
*Talaromyces* sp. XJA4	90.43 ± 0.01	>100	85.46 ± 0.3
*S. commune* XJA1	ND (not determined)	>100	89.8 ± 0.4
DOX	0.08 ± 0.2	0.1 ± 0.02	0.19 ± 0.01

**Table 6 microorganisms-08-00586-t006:** IC_50_ values of normal human embryonic kidney (HEK-293) cells and selectivity index (SI) values of antiproliferative crude extracts of *Aspergillus* sp. XJA6 and *A. terreus* XJA8.

Crude Extracts	IC_50_ (±SD, µg/mL)	SI (HT-29)	SI (Hela)
*Aspergillus* sp. XJA6	92.70.3 ± 1.4	nd	9.3
*A. terreus* XJA8	79.34 ± 0.7	13.8	nd

**Table 7 microorganisms-08-00586-t007:** Antioxidant activity of ethyl acetate crude extract from selected endophytic fungi.

Sample	IC_50_ (μg/mL)
*Aspergillus* sp. XJA6	No effect
*Talaromyces* sp. XJA4	171.78 ± 8.06
*A. terreus* XJA8	215.838 ± 7.25
*S. commune* XJA1	55.21 ± 0.3
Vitamin C	5.34 ± 0.42

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
