# Peer review of "Biological Activity of Endophytic Fungi from the Roots of the Medicinal Plant *Vernonia anthelmintica"

_microorganisms, 2020, doi:10.3390/microorganisms8040586_

Round 1

Reviewer 1 Report

The authors have improoved the text

I have only few formal suggestions

The graphical abstract was not added to the manuscript

Fig. 1  The text under the picture is not visible.

Line 99 change -  were tested for several biological activities

Line 196 Improve the sentence - All isolated fungal strains  cultivated on the PDB medium.....

Line 202 If the author are talking about sp. the first name of the microbe should be full written e.g. ???T. sp. XJA4  ????A. sp. XJA6 – check in the whole text

Author Response

RESPONSE TO REVIEWERS’ COMMENTS

REVIEWER #1: The authors have improved the text. I have only few formal suggestions.

Authors: Thanks for reviewer for these comment.

----------------------------------------------------------------------------------------

Comment 1: The graphical abstract was not added to the manuscript.

Reply: The graphical abstract is added to the revised manuscript.

----------------------------------------------------------------------------------------

Comment 2: Fig. 1  The text under the picture is not visible.

Reply: The text under the picture (Figure 1) have revised (we have increased font).

----------------------------------------------------------------------------------------

Comment 3: Line 99 change -  were tested for several biological activities.

Reply: Changed.

----------------------------------------------------------------------------------------

Comment 4: Line 196 Improve the sentence - All isolated fungal strains cultivated on the PDB medium.....

Reply: In the revised manuscript we have improved this sentence and printed in red color.

----------------------------------------------------------------------------------------

Comment 5: Line 202 If the author are talking about sp. the first name of the microbe should be full written e.g. ???T. sp. XJA4  ????A. sp. XJA6 – check in the whole text

Reply: Thank you for this comment. In the revised manuscript the first name of the microbes where sp. is now full written, and printed in red color.

Reviewer 2 Report

The manuscript entitled “Biological activity of endophytic fungi from roots of medicinal plant Vernonia anthelmintica” by Rustamova et al., shows the isolation of 15 fungi from the fresh root of a Chinese traditional medicinal plant, Vernonia anthelmintica. They also study the in vitro biological activities of ethyl acetate extracts obtained from these isolated fungi and performed the chemical characterization of the most active extract.

After major revision, they have partially improved the English language and style of the text and corrected most of the points suggested by reviewers, but it should be corrected a second time by a native speaker.

Some points still could be also improved:

Line 83-86 Authors must rewrite this paragraph to better explain the obtention of the root liquid. The obtention of dilutions and the composition of medium is also confusing.

Line 94 1 liter should be 1 L (correct units in different parts of the manuscript: min, µL, etc.)

Line 98 sulfoxide (DMSO) and then tested FOR (?) several biological activities.

The statistically significant differences are not marked in graphs by asterisks.

The discussion could be improved by relating the differences in biological activities and chemical composition of the different extracts. Which compounds must be responsible of those activities.

Author Response

RESPONSE TO REVIEWERS’ COMMENTS

REVIEWER #2: The manuscript entitled “Biological activity of endophytic fungi from roots of medicinal plant Vernonia anthelmintica” by Rustamova et al., shows the isolation of 15 fungi from the fresh root of a Chinese traditional medicinal plant, Vernonia anthelmintica. They also study the in vitro biological activities of ethyl acetate extracts obtained from these isolated fungi and performed the chemical characterization of the most active extract.

Authors: Thanks for reviewer for these comment.

----------------------------------------------------------------------------------------

Comment 1: After major revision, they have partially improved the English language and style of the text and corrected most of the points suggested by reviewers, but it should be corrected a second time by a native speaker.

Reply: Thank you for your comments. The revised manuscript has corrected a second time by a native speaker. All correction is now printed in red color in the revised manuscript.

----------------------------------------------------------------------------------------

Comment 2: Some points still could be also improved: Line 83-86 Authors must rewrite this paragraph to better explain the obtention of the root liquid. The obtention of dilutions and the composition of medium is also confusing.

Reply: In the revised manuscript this paragraph rewritten based on the reviewer suggestion. The production of dilutions also has improved in the revised manuscript and printed in red color.

----------------------------------------------------------------------------------------

Comment 3: Line 94 1 liter should be 1 L (correct units in different parts of the manuscript: min, µL, etc.)

Reply: Thank you for your useful comment. We have corrected all these units in the revised manuscript and printed in red color.

----------------------------------------------------------------------------------------

Comment 4: Line 98 sulfoxide (DMSO) and then tested FOR (?) several biological activities.

Reply: In the revised manuscript this sentence corrected as “The crude extract was dissolved in dimethyl sulfoxide (DMSO) and then were evaluated for their biological activities.”

----------------------------------------------------------------------------------------

Comment 5: The statistically significant differences are not marked in graphs by asterisks.

Reply: In the revised manuscript all figures have revised and now the statistically significant differences are marked in graphs by asterisks.

----------------------------------------------------------------------------------------

Comment 6: The discussion could be improved by relating the differences in biological activities and chemical composition of the different extracts. Which compounds must be responsible of those activities.

Reply: In our revised manuscript we improved discussion by the adding some sentences regarding chemical composition of the extract (lines 263-269, 312-314, 376-378). Present time we are finishing major work according separation of individually pure secondary metabolites (from ethyl acetate fraction), which we would like to evaluate their effect towards the biological targets. In our present work we have analyzed (chemical composition) only petroleum ether extract fraction by GC-MS and did not studied petroleum ether extract fraction toward the bioassays. However, by the adding some discussions along with citation related references follow your suggestion we improved discussion section. 

----------------------------------------------------------------------------------------

Round 2

Reviewer 2 Report

In spite of the efforts of the authors, English must still be improved. As examples, some sentences have been highlighted in the attached doccument. 

The authors should repeat properly the statistical analysis. There are asterisks everywhere, with no sense.

The number of replicates is lacking in some experiments.

Please find other suggestions in attached doccument.

Author Response

RESPONSE TO REVIEWERS' COMMENTS

REVIEWER 2:

Comment 1: In spite of the efforts of the authors, English must still be improved. As examples, some sentences have been highlighted in the attached doccument.

Author reply: In the revised manuscript we have improved English once again follow your suggestion; the native speakers in this field have corrected and revised our manuscript, in this regard we have acknowledged of their names in the manuscript (however corrections did not printed in red color, because native speakers revised whole manuscript). All sentences which highlighted in the attached document by reviewer also have corrected, and corrections and changes printed in red color.

----------------------------------------------------------------------------------------

Comment 2: The authors should repeat properly the statistical analysis. There are asterisks everywhere, with no sense.

Author reply: In the revised manuscript these points have corrected. The statistical analysis provided again using appropriate software.

----------------------------------------------------------------------------------------

Comment 3: The number of replicates is lacking in some experiments.

Author reply: Thank to reviewer for this useful comment. In the revised manuscript these points also have corrected and printed in red color.

----------------------------------------------------------------------------------------

Comment 4: Please find other suggestions in attached doccument.

Author reply: Thank to reviewer for these suggestion, we have corrected all suggestion that noted by reviewer in the revised manuscript.

This manuscript is a resubmission of an earlier submission. The following is a list of the peer review reports and author responses from that submission.

Round 1

Reviewer 1 Report

The article should be overwritten the English should be improved by the native speaker, lot of mistakes some words missing,  the discussion is poor, it is only a recapitulation of obtained results, Information in the discussion are very similar like these in the Introduction, no discussion about the chemical composition of the extracts and biological activity. It is not clear if the authors had assayed the obtained fractions as well or only the crude extracts? There was no suggestion, if the biological activity is dependent on some components or if it is the synergism of all the extracts components. Some of the information in the text are confusing and not clear written the whole text should be improved. The results should be better explained and the discussed.

Graphical abstract – only antimicrobial activity is mentioned,  it does not give the overview of the whole results that were obtained in this work.

Line 18 is after which is missing   Total 15 fungi were (add into the sentence)

Line21  if  it is not clear  to which species the Aspergillus XJA6 belongs  it should be  listed as Aspergillus sp.  or Aspergillus sp., isolate XJA6

Line43 terpenoids low case T (t) and comma(,) is missing after  terpenoids

Line 48 activ e is missing active

Line 50 acetylcholineesterase? What ? something is missing in the sentence

Line 51 fungus Bacillus niger ? Bacillus is not a fungus but bacteria,  Does really Bacillus niger exsists? Should it be not Aspergillus niger? Or Bacillus subtillis subsp. niger? If it is  bacteria do Bacillus sp. have secondary metabolites?

Line 53 Is the unit correct should it be really mg/ml , if it is so is not the concertation to high?

Line 58 the abbreviation PDB should be explained  (potato dextrose broth)?

Line 73 what did authors mean with the word ground? How were the root liquids obtained. Were the roots  mixed?

Line 76 potato  in what form it should be not potato extract

Line 77 chloramphenicol is not bactericide but bacteristatic it should be changed to antibacterial agent ( or for inhibition the growth of bacterial cells)

Line 88  after secondary metabolites were is missing

Line 92 all abbreviation in the text should be explained (PTP 1B, MTT, DPPH)

Line 100 why is the LB growth medium containing chloramphenicol?

Fig. 1  - Are the isolates included in the phylo-tree it is not clear (if not  these should be  included and maked for exaplme in bold)

Line 212 Fig. 1 should be replaced to Fig.2

If the authors will evaluate properly the antimicrobial activity, they should it assay the antimicrobial activity by dilution method and to evaluate the MIC values.

Line 232 Was the concentration of the metabolites really  uM, what molecular weight was used for getting the concentration of the extracts in molar concentration?

The whole section 3.4. - about the influence on melanine synthesis – authors are talking about the activity of  extracts not about activity of strains (it should be improved  in the text), I got totally lost in the numbers in parenthesis.

Section  3.5 the units after numbers are missing, the whole text in  this section does not give sense the sentences should be overwritten and checked by the native speaker For example :  line 262 XJA4show good inhibited on PTP-1B, line 265 Schizophyllum commune XJA1 extract inhibited antidiabetic activity moderate IC50 ????

Table.1  What does the cell free suspension mean? Were there the extracts assayed or not? The last row in the table, the unit shout be removed

Section 3.6 the same comment as to the section 3.4 Was the concentration of the metabolites really  uM, what molecular weight was used for the evaluation of the concentration?

Are authors still talking about the activity of extracts  or  about the activity of  strains? (it should be metioned in the text). More for the evaluation for cytotoxic  activity also an non - carcinogenic cell line should be evaluated because of the evaluation of general toxicity of the extracts.

Line 286 Table 2. Change the head of the table not MTT assay of fungal endophytes  but of ethyl acetate extracts obtained from fugal endophytes

line 298 Table3. Antioxidant activity of cell free suspension of endophytic fungi, Cell free or extracts???

 The chemical analysis should be placed in the text before the results of biological activities.

 Line 317  change to produce  to of

Line319 add are;  which are produced

Line324 add were;  were tested

Line 326 change terries – to terreus

Line 340 S. aureus the „s„ is missing

Reviewer 2 Report

The manuscript entitled “Biological activity of endophytic fungi from roots of medicinal plant Vernonia anthelmintica” by Rustamova et al., shows the isolation of 15 fungi from the fresh root of a Chinese traditional medicinal plant, Vernonia anthelmintica. They also study the in vitro biological activities of ethyl acetate extracts obtained from these isolated fungi and performed the chemical characterization of the most active extract.

Although the study is interesting, I found several problems. The authors should edit the English language and style of the text, as there are numerous grammar and typographical mistakes. Specifically, they should correct typographical mistakes in SI units and scientific nomenclature of fungi species and strains. Some corrections are also marked in the attached document.

The graphical abstract could be improved by replacing the histogram figure by other kind of graphic (letters are not readable).

The introduction should describe the traditional use of the selected plant to justify this study. It should be mentioned that the traditional anti-vitiligo drug is constituted by the fruit or the seeds of this plant.

Line 73. Authors should specify for how long they kept roots in sterile distilled water.

Line 224. Abbreviations (CA, SA, EC) should be explained in the legend.

It is not clear how the authors select these four fungi extracts, the screening results should be mentioned and could be suitable for supplementary figures.

Error bars and statistical analysis are lacking. It is necessary that authors include this information to compare the values obtained after the different treatments.

The relationship between the chemical composition of active fractions and their activities should be discussed.
